# Endophytism of *Lecanicillium* and *Akanthomyces*

**Rosario Nicoletti [1,2,\*] and Andrea Becchimanzi [2]**

[1] Council for Agricultural Research and Economics, Research Centre for Olive, Fruit and Citrus Crops, 81100 Caserta, Italy

[2] Department of Agricultural Sciences, University of Naples Federico II, 80055 Portici, Italy; andrea.becchimanzi@unina.it

[\*] Correspondence: rosario.nicoletti@crea.gov.it

**Abstract:** The rise of the holobiont concept confers a prominent importance to the endophytic associates of plants, particularly to species known to be able to exert a mutualistic role as defensive or growth-promoting agents. The finding that many entomopathogenic fungi are harbored within plant tissues and possess bioactive properties going beyond a merely anti-insectan effect has recently prompted a widespread investigational activity concerning their occurrence and functions in crops, in the aim of an applicative exploitation conforming to the paradigm of sustainable agriculture. The related aspects particularly referring to species of *Lecanicillium* and *Akanthomyces* (Sordariomycetes, Cordycipitaceae) are revised in this paper, also in light of recent and ongoing taxonomic reassessments.

**Keywords:** entomopathogens; endophytic fungi; crop protection; plant growth promotion; integrated pest management; bioactive compounds; Cordycipitaceae

## 1. Introduction

The great microbial diversity harbored in plants has just started being explored in light of a consolidated awareness that what we manage in the agricultural practice is actually the outcome of the combined expression of plant and microbial genes [1,2]. The symbiotic relationships between endophytic fungi and their host plants exteriorize in many ways, ranging from opportunistic saprophytism in senescent tissues, to latent pathogenicity disclosing after the impact of various stress factors, to genuine mutualistic interactions deriving from nutritional support and/or increased protection against pests and pathogens. The latter are particularly relevant for the holistic approach making its way in integrated pest management (IPM), considering the crop production system as a whole in the aim to contain rather than eradicate pests.

Within this conceptual rearrangement, the improvement of our knowledge on occurrence and functions of endophytic associates of plants is fundamental in view of their possible exploitation in sustainable agriculture. Endophytic entomopathogens are an important category of the plant microbiome, which is increasingly considered for applicative purposes. So far, the majority of investigations and reports concerning these organisms deal with *Beauveria bassiana* and *Metarhizium anisopliae*, with several fine reviews available in the literature [3,4]. This paper offers an overview on the current knowledge concerning endophytism in species of *Lecanicillium* and *Akanthomyces* (Sordariomycetes, Hypocreales, Cordycipitaceae).

## 2. Taxonomic Background

Until the early 2000s, these fungi were classified in the section *Prostrata* of the genus *Verticillium*, basically with reference to their imperfect stage producing verticillate conidiophores [5]. A few species best known for their parasitic behavior against arthropods, nematodes and/or fungi were

ascribed to this section, such as *V. chlamydosporium*, *V. lecanii* and *V. psalliotae*. Afterwards, the application of biomolecular techniques enabled to shed light on the phylogenetic relationships within this heterogeneous genus. Particularly, species within the section *Prostrata* were separated in a few unrelated genera, such as *Pochonia*, *Haptocillium*, *Simplicillium* and *Lecanicillium*, and their teleomorphs identified within the genera *Cordyceps* and *Torrubiella* [6]. The species *V. fungicola*, previously ascribed to the section *Albo-erecta* in the genus *Verticillium*, was later aggregated to *Lecanicillium* [7]. As a result of this fundamental revision, about fifteen *Lecanicillium* species were recognized, a few of which (*L. attenuatum*, *L. longisporum*, *L. muscarium*, *L. nodulosum* and *L. lecanii* s.str.) enucleated from the previously collective *V. lecanii*.

However, as it often happens in fungal taxonomy, such a sound rearrangement was not destined to persist. In fact the genus *Lecanicillium* was shown to be paraphyletic [8], and some species were moved to *Akanthomyces*, a pre-existing but overlooked genus including entomogenous species [9] (Table 1). At the same time, investigations in more or less peculiar ecological contexts brought to the description of novel taxa of both *Akanthomyces* and *Lecanicillium* [10,11], while some species ascribed to the latter genus, such as *L. uredinophilum* and *L. pissodis*, were shown to actually fit in the *A. lecanii* clade [12]. Following the dismissal of the dual nomenclature system for pleomorphic fungi, a more comprehensive revision of the whole family of the Cordycipitaceae is in progress. Particularly, rejection has been proposed for the genus name *Lecanicillium*, while some *Akanthomyces* species have in turn been moved to another genus (*Hevansia*) [13]. Hence, further adjustments concerning species still classified in *Lecanicillium* are to be expected.

**Table 1.** Nomenclatural correspondence of accepted *Lecanicillium*/*Akanthomyces* species with sequences of internal transcribed spacers of ribosomal DNA (rDNA-ITS) available in GenBank.

| Species Names * | | | ITS Sequence Used in |
|---|---|---|---|
| *Lecanicillium* | *Akanthomyces* | *Cordyceps/Torrubiella* | **Phylogenetic Analysis** |
| *L. acerosum* | | | NR11268 |
| *L. antillanum* | | | AJ292392 |
| *L. aphanocladii* | | | LT220701 |
| *L. aranearum* | *A. aranearum* | *T. alba* | AJ292464 |
| *L. araneicola* | | | AB378506 |
| *L. araneogenum* | *A. neoaraneogenus* | | NR161115 |
| *L. attenuatum* | *A. attenuatus* | | AJ292434 |
| *L. cauligalbarum* | | | MH730663 |
| *L. coprophilum* | | | MH177615 |
| *L. dimorphum* | | | AJ292429 |
| *L. flavidum* | | | EF641877 |
| *L. fungicola* var. *aleophilum* | | | NR111064 |
| *L. fungicola* var. *fungicola* | | | NR119653 |
| *L. fusisporum* | | | AJ292428 |
| *L. kalimantanense* | | | AB360356 |
| *L. lecanii* | *A. lecanii* | *C. confragosa* | AJ292383 |
| *L. longisporum* | *A. dipterigenus* | | AJ292385 |
| *L. muscarium* | *A. muscarius* | | NR111096 |
| *L. nodulosum* | *Akanthomyces* sp. | | EF513012 |
| *L. primulinum* | | | NR119418 |
| *L. psalliotae* | | | AJ292389 |
| *L. restrictum* | | | LT548279 |
| *L. sabanense* | *A. sabanensis* | | KC633232 |
| *L. subprimulinum* | | | MG585314 |
| *L. tenuipes* | | | AJ292391 |
| *L. testudineum* | | | LT548278 |
| *L. uredinophilum* | *Akanthomyces* sp. | | MG948305 |
| *L. wallacei* | | *T. wallacei* | NR111267 |
| *Lecanicillium* sp. | | *C. militaris* | AF153264 |
| | *A. aculeatus* | | KC519371 |

| | | |
|---|---|---|
| ***A. coccidioperitheciatus*** | *C. coccidioperitheciata* | JN049865 |
| ***A. kanyawimiae*** | | MF140751 |
| *A. sphingum* | ***C. sphingum*** | AY245641 |
| ***A. sulphureus*** | *Torrubiella* sp. | MF140756 |
| ***A. thailandicus*** | *Torrubiella* sp. | MF140755 |
| ***A. tuberculatus*** | *C. tuberculata* | JN049830 |
| ***A. waltergamsii*** | | MF140747 |

\* The currently used species names as inferred from the Mycobank database [14] are reported in bold.

## 3. Occurrence

The number of reports concerning endophytic isolates of *Lecanicillium* and *Akanthomyces* has increased in recent years. This is due not only to the several taxonomic reassessments introducing new species, but also to the easier access to techniques and databases for DNA sequencing, which in most instances enable one to overcome the intrinsic difficulties of morphological identification. However, more prompts have probably resulted by the awareness of the basic role that endophytic fungi play on plant fitness, introducing applicative perspectives for investigations in the field. For the above genera, literature shows a prevalence of findings concerning natural phytocoenoses (Table 2) over those inherent crops (Table 3); even more so considering that the latter series includes a few cases of endophytic colonization resulting after artificial inoculation in experimental work. Basically connected with the issue of ecosystem simplification characterizing the agricultural contexts, such a difference emphasizes the opportunity to recover the functional role of this component of the plant holobiont in view of improving crop performances.

**Table 2.** Endophytic occurrence of *Lecanicillium*/*Akanthomyces* in wild contexts.

| Species | Host plant | Country | ITS Sequence √ | Reference |
|---|---|---|---|---|
| *A. attenuatus* | *Astrocaryum sciophilum* | French Guyana | MK279520 | [15] |
| | Conifer plant | China | MN908945 | GenBank |
| | *Symplocarpus foetidus* | Canada | KC916681 | [16] |
| *A. lecanii* | *Ammophila arenaria* | Spain | - | [17] |
| | *Dactylis glomerata* | Spain | AM262369 | [18] |
| | *Deschampsia flexuosa* | Finland | KJ529005 | [19] |
| | *Elymus farctus* | Spain | AM924163 | [17] |
| | *Laretia acaulis* | Chile | - | [20] |
| | *Pinus sylvestris* | Italy | KJ093501 | [21] |
| | *Pinus sylvestris* | Poland | - | [22] |
| | *Shorea thumbuggaia* | India | KJ542654 | GenBank |
| | *Taxus baccata* | Iran | KF573987 | [23] |
| *A. muscarius* | *Acer campestre* | Italy | MT230457 | This paper |
| | *Laurus nobilis* | Italy | - | [24] |
| | *Myrtus communis* | Italy | MT230435 | This paper |
| | *Nypa fruticans* | Thailand | MH497223 | [25] |
| | *Quercus robur* | Italy | MT230463 | This paper |
| *Akanthomyces* sp. * | *Arctostaphylos uva-ursi* | Switzerland | - | [26] |
| | *Carpinus caroliniana* | USA | - | [27] |
| *L. aphanocladii* | *Ageratina adenophora* | China | MK304090 MK304173 MK304418 | [28] |
| | *Hemidesmus indicus* | India | MH594215 | [29] |
| | *Huperzia serrata* | China | KP689216 KP689173 | [30] |
| | *Picea mariana* | Canada | - | [31] |
| *L. fungicola* | *Phragmites australis* | Korea | KP017880 | [32] |
| *L. kalimantanense* | *Zingiber officinale* | Indonesia | - | [33] |
| *L. psalliotae* | *Cerastium fischerianum* | Korea | JX238776 | [34] |
| | *Coix lachryma-jobi* | China | KJ572167 | GenBank |

|  | Magnolia officinalis | China |  | GenBank |
|---|---|---|---|---|
|  | Phoradendron perrottettii | Brazil | - | [35] |
|  | Pinus radiata | New Zealand | - | [36] |
|  | Sedum oryzifolium | Korea | KU556134 | [37] |
|  | Tapirira guianensis | Brazil | - | [35] |
|  | Triticum dicoccoides | Israel | - | [38] |
| Lecanicillium sp. | Artocarpus lacucha | India | MH700423 MH700428 | GenBank |
|  | Bupleurum chinense | China | MG561939 | GenBank |
|  | Huperzia serrata | China | KM513600 | [30] |
|  | Liparis japonica | China | KT719186 KT719187 KT719188 KT719189 KT719192 | GenBank |
|  | Micrandra spruceana | Peru | MH267985 | [39] |
|  | Microthlaspi perfoliatum | Greece | KT269776 | [40] |
|  | Quassia indica | India | MH910098 | GenBank |
|  | Sandwithia guyanensis | French Guyana | MN514023 | [41] |
|  | Theobroma gileri | Ecuador | - | [42] |

√ Missing ITS accession number implies identification based on morphological characters only, or without depositing the ITS sequence. * These strains were originally identified as *Verticillium lecanii*.

**Table 3.** Endophytic occurrence of *Lecanicillium/Akanthomyces* in crops.

| Species | Host plant | Country | ITS Sequence √ | Reference |
|---|---|---|---|---|
| A. attenuatus | Brachiaria sp. | Kenya | KU574698 | [43] |
|  | Salvia miltiorrhiza | China | JX406555 | GenBank |
| A. lecanii | Cucurbita maxima | Australia | - | [44] |
|  | Gossypium hirsutum | Australia | - | [45] |
|  | Gossypium hirsutum | Brazil | - | [46] |
|  | Gossypium hirsutum | Texas, USA | KP407570 | [47] |
|  | Solanum lycopersicum | Australia | - | [44] |
|  | Phaseolus vulgaris | Australia | - | [44] |
|  | Phaseolus vulgaris | China | - | [48] |
|  | Pistacia vera | Iran | MF000354 | [49] |
|  | Triticum aestivum | Australia | - | [44] |
|  | Vitis vinifera | Spain | - | [50] |
|  | Zea mays | Australia | - | [44] |
| A. muscarius | Brassica oleracea | New Zealand | - | [51] |
|  | Cucumis sativus | Canada | - | [52] |
|  | Cucumis sativus | Japan | - | [53] |
|  | Prunus cerasus | Iran | KY472303 | [54] |
| L. aphanocladii | Zea mays | Slovenia | - | [55] |
| L. dimorphum | Phoenix dactylifera | Spain | - | [56] |
| L. psalliotae | Phoenix dactylifera | Spain | - | [56] |
| Lecanicillium sp. | Citrus limon | Iran | MN448344 | GenBank |
|  | Vitis vinifera | China | MT123107 | GenBank |
|  | Zea mays | India | - | [57] |

√ Missing ITS accession number implies identification based on morphological characters only, or without depositing the ITS sequence.

Overall, Tables 2 and 3 include 65 citations of endophytic strains belonging to these two genera as a result of a search considering literature in the field and the GenBank database. A widespread capacity to colonize plants from heterogeneous ecological contexts is evident considering that these citations refer to 54 species belonging to 35 botanical families. With 10 species Poaceae is the most represented family, followed by Arecaceae and Pinaceae with three species each, and

Anacardiaceae, Apiaceae, Brassicaceae, Cucurbitaceae, Euphorbiaceae and Malvaceae with two species. The rest of the families (Apocynaceae, Araceae, Asteraceae, Betulaceae, Caryophyllaceae, Crassulaceae, Dipterocarpaceae, Ericaceae, Fabaceae, Fagaceae, Lamiaceae, Lauraceae, Lycopodiaceae, Magnoliaceae, Moraceae, Myrtaceae, Orchidaceae, Rosaceae, Rutaceae, Santalaceae, Sapindaceae, Simaroubaceae, Solanaceae, Taxaceae, Vitaceae and Zingiberaceae) are represented by a single species.

Such a variety of hosts seems to contrast any hypothesis of host specialization, and is rather indicative of a possible tendency to spread horizontally within the phytocoenoses. In this respect, the recovery of *A. muscarius* from four woody species (*Acer campestre*, *Laurus nobilis*, *Quercus robur* and *Myrtus communis* in two separate stands) at the Astroni Nature Reserve near Napoli, Italy ([24] and in this paper), appears to support this ability, which may as well imply a permanent functional role in natural ecosystems. On the other hand, indications of a constant association with crop species could be favorable for possible applications in IPM. The limited available data only support preliminary clues in the case of cotton (*Gossypium hirsutum*) where, considering the economic impact of insect pests, the endophytic occurrence of strains of *A. lecanii* reported from distant countries such as Australia, Brazil and the United States might deserve further attention.

### 3.1. Phylogenetic Relationships of Endophytic Strains

In the evolving taxonomic scheme outlined above, the endophytic isolates provisionally classified as *Lecanicillium* sp. are to be further considered for a more definite taxonomic assignment. In this perspective, we propose a phylogenetic analysis (Figure 1) considering strains whose sequences of internal transcribed spacers of ribosomal DNA (rDNA-ITS) are deposited in GenBank (Tables 2 and 3), along with official reference strains for the currently accepted species of *Lecanicillium* and *Akanthomyces* (Table 1).

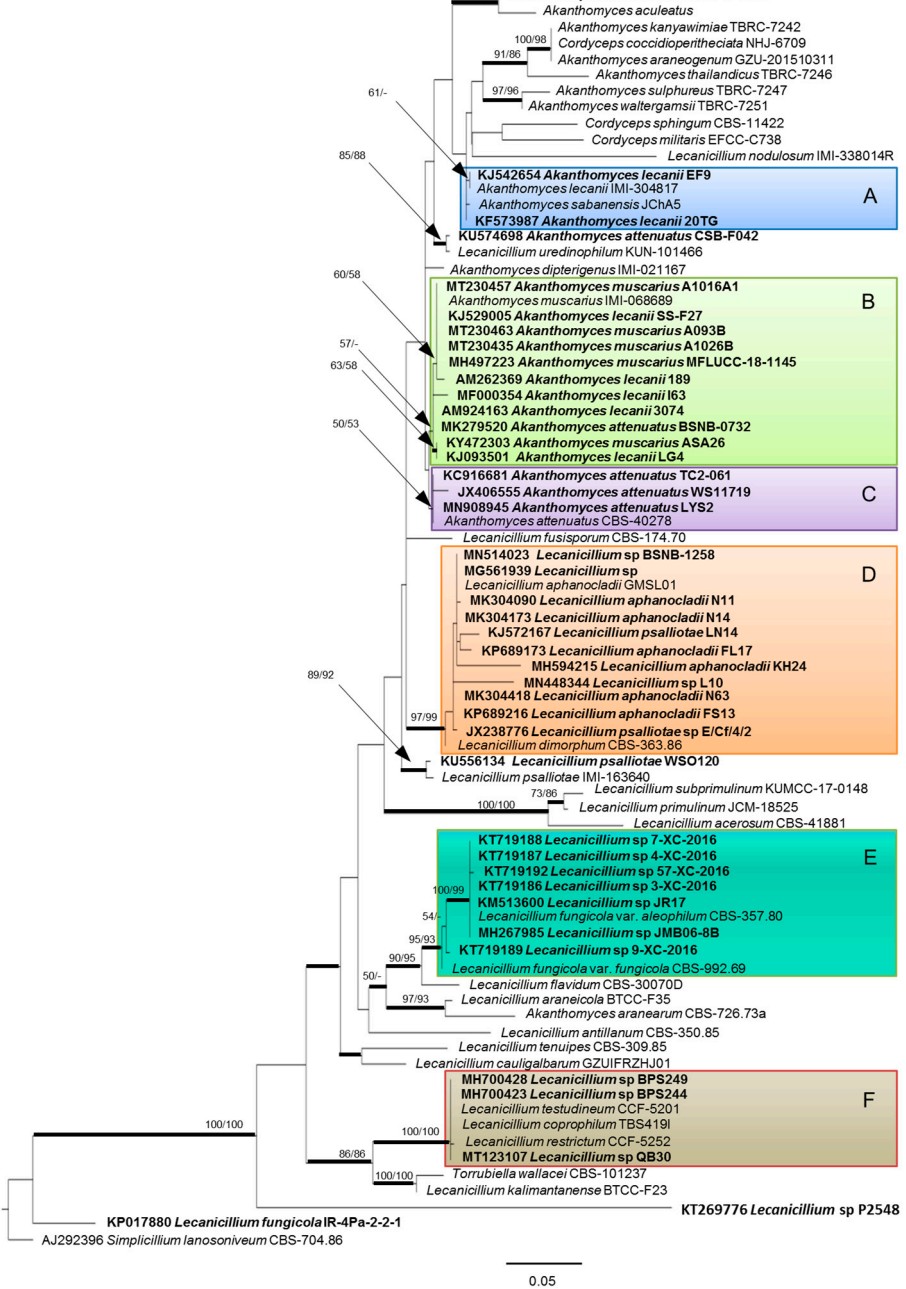

**Figure 1.** Phylogenetic tree based on maximum likelihood (ML) analysis of the rDNA-ITS sequences deposited in GenBank for the known species (Table 1) and the endophytic strains of *Lecanicillium* and *Akanthomyces* (in bold, Tables 2 and 3). Multiple sequence alignment comprised 592 nucleotide positions, including gaps. The analysis was carried out using RAxML software (version 8.2.12; https://cme.h-its.org/exelixis/web/software/raxml) for ML, PAUP (version 4.0a166; https://paup.phylosolutions.com) for maximum parsimony (MP), and MrBayes (version 3.2.7a; https://nbisweden.github.io/MrBayes/download.html) for Bayesian analysis. Phylogenetic tree was drawn using FigTree software (version 1.4.4; http://tree.bio.ed.ac.uk/software/figtree). Details and complete references are specified in a recent paper [58]. Bootstrap support values ≥60% for ML and MP are presented above branches as follows: ML/MP, bootstrap support values <50% are marked with '-'. Branches in bold are supported by Bayesian analysis (posterior probability ≥95%). *Simplicillium lanosoniveum* CBS 704.86 (GenBank: AJ292396) was used as outgroup reference. Main clades are indicated by colored boxes A, B, C, D, E and F.

Although more DNA sequences, such as the translation elongation factor 1 alpha (*TEF*) and RNA polymerase II largest subunits 1 (*RPB1*) and 2 (*RPB2*), are considered in taxonomic assessments concerning genera in the Cordycipitaceae [12,13,25,59], provisional identification of isolates recovered in the course of biodiversity studies is routinely done on account of ITS. Therefore only these kinds of sequences are usually deposited in GenBank for such strains, representing the only possible marker available for phylogenetic reconstructions.

In the absence of opportunities for a direct examination of these isolates, the phylogenetic tree proposed in Figure 1 provides an indication for their provisional assimilation to any of the accepted taxa in the genera *Lecanicillium* and *Akanthomyces*. A major cluster in the upper part of the tree includes the type strains of the species of *Cordyceps*, *Akanthomyces* (except *A. aranearum*), and of *L. nodulosum* and *L. uredinophilum*, which are also credited for ascription to *Akanthomyces*, along with all the endophytic strains ascribed to the species *A. lecanii*, *A. muscarius* and *A. attenuatus* (clades A, B and C, respectively). However, just two out of seven endophytic isolates ascribed to *A. lecanii* are next to the type strain of this species, while five more isolates rather group with *A. muscarius*. Confirming evidence from previous phylogenetic analyses [25,59], *A. attenuatus* is very close to *A. muscarius*, but an isolate from the palm *Astrocaryum sciophilum* is displaced in clade B. Another isolate from *Brachiaria* sp. reported as *A. attenuatus* is more distant, having *L. uredinophilum* as the closest relative. While these remarks cannot be taken as an evidence of a more common endophytic occurrence of *A. muscarius*, they represent an indication that at least some isolates of this species might have been misidentified as *A. lecanii*. This is not surprising, considering that a previous study pointed out the difficulty of resolving species ascription of strains previously ascribed to *V. lecanii* by using ITS sequences only [60].

Interestingly, no endophytic isolates provisionally identified as *Lecanicillium* sp. belong to the above major *Akanthomyces* cluster. Three of them are part of clade D, corresponding to the species *L. aphanocladii*, which also includes two strains identified as *L. psalliotae*. This is acceptable since these species and *L. dimorphum* have been reported in a close phylogenetic relationship in previous analyses [6,59]. However, *L. psalliotae* seems somehow problematic with reference to the resolution power of ITS, considering that it was reported as the closest relative (99.65% sequence identity at 100% query cover) of another isolate from *Microthlaspi perfoliatum* [40], which is in a quite distant position in our phylogenetic tree.

As many as seven unidentified strains cluster with *L. fungicola*, prevalently with the type strain of var. *aleophilum* (clade E), indicating a relevant endophytic occurrence of this species, which was not recognized so far. Another isolate reported as *L. fungicola* [32], deserves a more careful consideration with reference to its basal placement. In fact, BLAST search in GenBank indicated a 100% identity with ten strains of this species and several strains of the unrelated *Simplicillium aogashimaense*. The latter was characterized in 2013 with the support of a phylogenetic analysis based on ITS only, which anyway showed a consistent distance from *L. fungicola* [61]. Quite meaningfully, in our analysis the isolate in question was placed in proximity to the outgroup (*Simplicillium lanosoniveum*) on which our tree was rooted. Considering that sequences of six out of this group of ten *L. fungicola* strains were deposited in GenBank before 2013, it is quite possible that original misidentification of those that might rather have been *Simplicillium* strains could have determined the incorrect assignment of the more recent isolates.

Finally, three isolates (two Indian from *Artocarpus lacucha* and one Chinese from *Vitis vinifera*) are grouped in clade F together with the type strains of the recently described *L. coprophilum* [11], *L. restrictum* and *L. testudineum* [62]. A BLAST search in the GenBank database shows the first species as the closest relative, with 100% and 99.81% ITS sequence identity for the Chinese and the Indian isolates, respectively.

## 4. Implications in Crop Protection

As introduced above, so far there are few observations concerning the effects of endophytic strains of *Lecanicillium* and *Akanthomyces* in crops. Within the limited data available so far, cotton stands out for remarks on the endophytic occurrence of *A. lecanii* from independent cropping areas.

In Australia an endophytic isolate was shown to be able to colonize cotton plants ensuring protection against the cotton aphid (*Aphis gossypii*) after artificial inoculation. Besides evidence from direct microscopic examination, the ability to colonize plant tissues was confirmed by re-isolation from leaves of the treated plants, which was successful up to 35 days after inoculation. This persistence can be taken as an indication of an endophytic life strategy, considering that endophytic colonization enables the fungus to become resident in a stable and nutritious insect-attracting environment. High humidity enhanced colonization of both plants and aphids; this expected effect is relevant for the management of the cotton aphid, which is most commonly found in the lower canopy, where humidity is high and the fungus is more protected against the adverse effects of UV radiation from sun [63]. Moreover, contact with conidia of *A. lecanii* significantly reduced the rate and period of reproduction of *A. gossypii*. The culture filtrate of the fungus significantly increased mortality and reduced reproduction, while feeding-choice experiments indicated that the aphids might be able to detect the fungal metabolites. The ethyl acetate and methanolic fractions of culture filtrate and mycelia also caused significant mortality and reduced fecundity [64]. Besides cotton, the same strain displayed the ability to colonize plants of wheat, corn, tomato, bean and pumpkin after artificial inoculation of leaves, while soil inoculation was ineffective [44].

Additional reports from cotton come from Texas [47] and Brazil, where the endophytic occurrence of *A. lecanii* was detected in leaves and roots of both normal and *Bt*-transgenic plants [46]. Although no aspects concerning interactions with pests were evaluated in these cases, it is meaningful that several strains of *A. lecanii* were recovered in each of these three contexts, indicating a possible common association of this species with cotton, which deserves to be more thoroughly verified.

The adaptation of *A. lecanii* to exert entomopathogenicity in association with plants is well attested by the finding that the fungus responds to volatile compounds produced by the plant during insect feeding. Particularly, in a model based on thale cress (*Arabidopsis thaliana*) and the mustard aphid (*Lipaphis erysimi*), compounds such as methyl salicylate and menthol were found to promote spore germination and pathogenicity of the fungus [65,66].

Besides aphids, protective effects after systemic colonization have been demonstrated against the red spider mite (*Tetranychus urticae*) in bean plants. In this case a strain of *A. lecanii* was reported to spread within the plant tissues after artificial inoculation of seeds, promoting growth and impairing survival and fecundity of the mites. These effects were even carried over the following generation of mites fed on fresh plants [48].

Pathogenicity of *A. lecanii* against a wide array of noxious arthropods is integrated by antagonism towards plant pathogenic fungi. In addition to a general antifungal activity demonstrated in vitro against polyphagous species such as *Sclerotinia sclerotiorum*, *Rhizoctonia solani* and *Aspergillus flavus* [49], possible exploitation of this double functionality has been conceived on several crops, such as coffee where *A. lecanii* behaves as both a parasite of the leaf rust (*Hemileia vastatrix*) and a pathogen of the green scale (*Coccus viridis*) [67]. The same role can be considered in crops where powdery mildews can represent a major phytosanitary problem, such as cucurbits [68,69].

Moreover, antifungal effects could derive from stimulation of the plant defense response, as reported for an endophytic strain able to promote such reaction against *Pythium ultimum* in transformed cucumber plants [52]. Additional experimental evidence in this regard is provided by observations carried out on the date palm (*Phoenix dactylifera*) where the inoculation of endophytic strains of *L. dimorphum* and *L.* cf. *psalliotae,* previously reported for entomopathogenicity against the red palm scale (*Phoenicococcus marlatti*) [56], induced proteins involved in plant defense or stress response. Proteins related with photosynthesis and energy metabolism were also upregulated, along with accumulation of a heavy chain myosin-like protein [70].

The concurrent role against plant pests and pathogens is known to operate for other *Lecanicillium* and *Akanthomyces* species, and for non-endophytic strains of various origin, as more in detail discussed in dedicated papers [71,72]. The need to combat multiple adversities has also prompted the evaluation of a possible combined use of these fungi with chemical pesticides. In this respect, it has been observed that the spread of *A. lecanii* in plant tissues is not affected by treatments

with insecticides belonging to several classes [73]. Moreover, substantial safety of insecticides has been reported in in vitro assays carried out on *A. muscarius*, while several herbicides and fungicides were responsible for negative effects or even suppression of mycelial growth [74]. For the latter species, in vivo observations on the sweet potato whitefly (*Bemisia tabaci*) demonstrated the positive effects of association with chemical insecticides in view of reducing their use, particularly in the greenhouse [75]. Again with reference to application of *A. muscarius* for the control of *B. tabaci*, it is worth mentioning the synergistic effects resulting in combined treatments with matrine, a plant-derived quinolizidine alkaloid [76].

In addition to the indirect side effects deriving from protection against biotic and abiotic adversities, many endophytes have been reported to promote plant growth through essentially two mechanisms; that is the release of plant hormones, or the improvement of nutritional conditions. Of course, strains possessing both properties are likely to contribute in an additive manner, as observed for an isolate of *L. psalliotae* from cardamom (*Elettaria cardamomum*). Besides producing indole-3-acetic acid, this strain enhanced chlorophyll content of leaves as a likely result of release of siderophores, and increased availability of zinc and inorganic phosphate by promoting their solubilization [77]. Release of siderophore has also been reported for an endophytic isolate of *A. lecanii* from *Pistacia vera* [49].

## 5. Biochemical Factors Involved in the Tritrophic Interaction with Plants and Pests

It has been previously introduced that, at least in part, the antagonistic/pathogenic ability by *Lecanicillium* and *Akanthomyces* strains is mediated by biochemical factors, such as enzymes and secondary metabolites. Endophytic fungi are regarded as a goldmine of undescribed chemodiversity, and even diffusely reported as capable to synthesize bioactive products originally characterized from their host plants [78]. Although it is quite reasonable that they exploit this biosynthetic potential in the natural environment, more rigid opinions occasionally question a real role by these compounds until their production is demonstrated in plants. Pending a solution of this diatribe through the development of methods for ascertaining their effective release and bioactivity in plant tissues, so far research in the field has disclosed interesting properties by species of *Lecanicillium* and *Akanthomyces*, too.

The first metabolomic studies concerning these fungi were carried out with strains of *V. lecanii* before the taxonomic revision. Two isolates were found to produce 2,6-dimethoxy-*p*-benzoquinone, phenylalanine anhydride, aphidicolin and dipicolinic acid, with the latter showing insecticidal effects in bioassays on the blowfly *Calliphora erythrocephala* [79]. Afterwards, two more triterpenoid carboxylic acids with alleged insecticidal properties were reported from the same source [80]. Incompletely identified toxic products, possibly phospholipids, were extracted from another strain showing activity against *B. tabaci*, the western flower thrips (*Frankliniella occidentalis*) and a few aphid species [81]. Anti-insectan effects against the corn earworm (*Helicoverpa zea*) were later reported for vertilecanin A, the most abundant component in a group of five new phenopicolinic acid analogues [82]. Moreover, two structurally unidentified products were extracted from two Chinese strains, displaying toxic, ovicidal and antifeedant properties against *B. tabaci* [83]. Finally, the novel indolosesquiterpenes lecanindoles A-D, with quite peculiar structures and bioactivities, were characterized from another aphidiculous strain [84].

Later on more strains were found to produce novel compounds without a direct connection with their entomopathogenicity. Two inactive aromadendrane sesquiterpenes, inonotins M and N, were extracted from a strain of *L. psalliotae* [85]. An unidentified *Lecanicillium* sp. was reported to produce lecanicillolide [86], and lecanicillones A-C, three unusual dimeric spiciferones with an acyclobutane ring displaying moderate cytotoxic effects [87]. More interesting inhibitory effects on tube formation by endothelial cells, implying antiangiogenic properties, were reported for the decalin polyketide 11-norbetaenone, from a strain of *L. antillanum* [88].

Besides novel compounds, investigations on these fungi have also disclosed the production of well-known bioactive metabolites. A strain of *L. psalliotae* was found to produce oosporein, a common product of *Beauveria* spp., which displayed strong inhibitory activity against the potato late

blight fungus (*Phytophthora infestans*) [89]. Likewise, several cyclic depsipeptides have been reported from miscellaneous isolates. The list includes eight destruxin analogues, well-known secondary metabolites of *M. anisopliae*, by strain KV71 of *L. longisporum* (the active principle of the mycoinsecticide Vertalec) [90]; bassianolide, previously reported from *B. bassiana*, from *A. lecanii* [91], and the antifungal verlamelins A-B, previously known from *Simplicillium lamellicola*, from an unidentified *Lecanicillium* strain [92]. Finally, stephensiolides C, D, F, G and I, originally characterized from a gram-negative bacterium (*Serratia* sp.) symbiotic with a mosquito (*Anopheles stephensi*), have been recently detected in the culture extract of an endophytic *Lecanicillium* sp. as the bioactive principles responsible for antibacterial activity against methicillin-resistant *S. aureus* [41]. Inhibitory properties against the same bacteria, along with cytotoxicity on human lung fibroblast cells, were ascribed to cyclic depsipeptides contained in the culture extracts of a strain of *A. attenuatus* [15].

Antibiotic effects against *S. aureus* were also reported for akanthomycin, extracted from cultures of *Akanthomyces gracilis* together with the closely related pyridine alkaloids 8-methylpyridoxatin and cordypyridone C [93]. Additional findings from *Akanthomyces novoguineensis* concerning the akanthopyrones [94], akanthol, akanthozine, butanamide and oxodiazanone derivatives [95] are not to be further considered in this review by reason that this species is now classified in the genus *Hevansia* [14].

This concise analysis of the pertinent literature, mostly made of independent or occasional findings, highlights the importance of carrying out more systematic work on the metabolomics of members of *Lecanicillium* and *Akanthomyces*. In fact, a thorough revision could ascertain whether some compounds eventually represent biochemical markers for selected species, and which products are effectively associated with the expression of pathogenicity towards insects, nematodes and spiders, as well as with antagonism/mycoparasitism against plant pathogens. In this respect, an interesting hypothesis has been advanced concerning the above-mentioned dipicolinic acid, which is known to act as a prophenoloxidase inhibitor and an immunosuppressive agent in insects. After its concomitant detection as a product of several entomopathogenic species belonging to the Hypocreales, including *A. muscarius*, it has been advanced that the acquired ability to synthesize this compound might have shaped evolution of these fungi from mere plant associates to the more specialized lifestyle as arthropod pathogens [96].

Literature on enzyme production by endophytic strains of *Lecanicillium* and *Akanthomyces* is more limited. Chitinolytic enzymes are not only necessary to these fungi to penetrate cuticle of insects, nematodes or spiders, but they are also involved in the activation of the disease response by the plant and induction of systemic resistance [97–99]. The same function may also be played by other enzyme complexes, such as proteases and β-glucanases, which are known to integrate the enzymatic profile of many endophytes [100–102]. Besides directly affecting survival and fecundity of the green peach aphid (*Myzus persicae*) in a concentration-dependent manner, a protein characterized from a strain of *A. lecanii* was found to concomitantly induce upregulation in tomato plants of genes associated to the salycilate and jasmonate pathways, which are involved in the systemic response to biotic stress [103].

## 6. Future Perspectives

As a likely heritage of old investigational schemes, there is a cultural propensity in research projects and reports to refer to plant-associated microorganisms within the boundaries of functional categories. However, it is increasingly evident that many endophytic fungi are eclectic and possess a multifaceted connotation enabling them to perform several more or less interconnected beneficial roles in the symbiotic relationship with their host plants. Such a revised concept particularly applies to species of *Lecanicillium* and *Akanthomyces*, which should not be merely regarded as entomopathogenic fungi anymore.

Strains of the species *A. lecanii, A. muscarius, A. attenuatus* and *A. longisporus* are already used as the active ingredients of several mycoinsecticides [72]. Although their inclusion in IPM appears to be an obvious approach, a more efficient employment should be pursued in light of the body of evidence disclosed by recent experimental work that, besides killing pests as a result of inundative

treatments, the endophytic establishment of these fungi may have further relevance on plant fitness. That is a clear antagonistic role against plant pathogens, the capacity to stimulate plant defense reactions and various plant growth promoting effects.

These valuable properties, shared with other species of *Lecanicillium* and *Akanthomyces*, make it advisable to carry out extensive investigations in crops to verify the natural endophytic occurrence, and to increase our knowledge on ecology of these fungi. Particularly relevant is gathering additional information on the production in plants of the biochemical factors, which possibly play a role in regulating the tritrophic relationship with the host and its pests/pathogens.

At the same time it is fundamental to assess whether their endophytic establishment is possible following artificial introduction. In this respect, inoculation methods (foliar spraying, soil drenching, seed soaking, and injections) are crucial for an enduring survival within plant tissues, and their compliance should be more accurately evaluated [104]. Particularly in crops where these fungi can exert a positive impact, additional observations are appropriate to verify whether their distribution pattern is localized or systemic. Actually, a great challenge for considering endophytic fungi as a strategy in plant protection is to manage their reproducible introduction into crops, and to predict the outcome. As well, the effectiveness of this attractive phytosanitary tool needs to be proven in the field to stimulate growers to adopt it in view of gaining clear economic benefits.

**Author Contributions:** Conceptualization, R.N.; phylogenetic analysis, A.B.; data curation, A.B.; writing—original draft preparation, R.N.; writing—review and editing, R.N. and A.B. All authors have read and agreed to the published version of the manuscript.

**Funding:** This research received no external funding.

**Conflicts of Interest:** The authors declare no conflict of interest.

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
