# Peer review of "Endophytism of Lecanicillium and Akanthomyces"

_agriculture, doi:10.3390/agriculture10060205_

Round 1

Reviewer 1 Report

This review manuscript is interesting and I think will make a relevant contribution to the field. The language in some places could be simplified for better readability.

I have a few minor comments and suggestions for revisions:

L76-80: Please clarify the sentence starting with "Other than deriving...", it may need to be broken down into several shorter sentences to make more sense.

L81 and L83: Tables 2 & 3, did the references that are missing ITS accession numbers include other nucleotide sequence accessions? If not, were the fungi identified/confirmed in those studies only by morphology? It might be worth including these details, perhaps even as a footnote to the table(s) since endophyte misidentification has been known to happen.

L100: I suggest to replace the word 'aptitude' in the the text, wherever written, for either ability or capability.

Section 3.1.
L109: You haven't conducted a phylogenetic analysis, it is a gene tree as it is based on ITS alone. That doesn't render the result irrelevant to the overall review but it should be discussed perhaps as a preliminary analysis that requires further validation and support by at least including/combining mitochondrial sequences (which may evolve at a different rate to nuclear genes), to verify if the indicated clades cluster comparatively.

L110-111: You also need to provide here the methodology on how the data analysis was performed. How long was the ITS sequence used in the multiple sequence alignment?

L115: Although you cite a prior paper in the figure caption (58), I think you also need to provide the details on how the analysis was performed in the text because they are not the same. The other manuscript provides an analysis based on two loci as it includes TEF1-α. 

L121-126: I appreicate that other nuclear/mitochondrial sequences may not be available in Genbank (or other database?) for all species presented here but that doesn't mean that this can be called a phylogenetic analysis. Either acquire the isolates and sequence them, and then include the extra loci in the phylogeny or state that this will need to be done in a future study to verify your gene-tree presented here.

L204: The meaning of this sentence is unclear, please revise.

L207: Not sure what you mean by "ambivalent aptitude...envisaged". Use 'observed' or 'reported' instead of envisaged. Ambivalent in english means 'unsure' or uncertain, I don't think this is what you mean.

L222: Use 'combat' instead of 'contrast'.

L234-235: Endophytes do not always provide a benefit to the plant, so please consider revising this sentence to include this point so you don't overstate their potential.

Author Response

Apart of #3, who seems not to have even read the manuscript, we wish to thank reviewers for their careful revision and suggestions for improving our paper. Adjustments introduced in the new version and responses to the reviewers’ comments are reported below.

Reviewer 1:

This review manuscript is interesting and I think will make a relevant contribution to the field. The language in some places could be simplified for better readability.

I have a few minor comments and suggestions for revisions:

L76-80: Please clarify the sentence starting with "Other than deriving...", it may need to be broken down into several shorter sentences to make more sense.

This sentence has been shortened.

L81 and L83: Tables 2 & 3, did the references that are missing ITS accession numbers include other nucleotide sequence accessions? If not, were the fungi identified/confirmed in those studies only by morphology? It might be worth including these details, perhaps even as a footnote to the table(s) since endophyte misidentification has been known to happen.

Entries where ITS accession number is missing were either identified through morphological characters only, or without depositing an ITS sequence. This is been specified in the table footnotes.

L100: I suggest to replace the word 'aptitude' in the the text, wherever written, for either ability or capability.

The word ‘aptitude’ has been changed throughout the text.

Section 3.1.
L109: You haven't conducted a phylogenetic analysis, it is a gene tree as it is based on ITS alone. That doesn't render the result irrelevant to the overall review but it should be discussed perhaps as a preliminary analysis that requires further validation and support by at least including/combining mitochondrial sequences (which may evolve at a different rate to nuclear genes), to verify if the indicated clades cluster comparatively.

Actually, our analysis only intended to verify if any of the strains reported as ‘Lecanicillium sp.’ could be ascribed to any of the recognized Lecanicillium/Akanthomyces species. It had no purpose to interfere in the taxonomic assessments/reassessments which are in progress concerning these fungi. We do agree that it is partial and preliminary, still it suits to our intentions. The tree results from 3 different phylogenetic analysis (Maximum likelihood, Maximum parsimony and Bayesian), and even though it does not presume to represent a taxonomic insight, we consider that the term ‘phylogenetic’ is correctly used. In this respect, previous phylogenetic analysis based on ITS only have been previously published for several fungi, such as Fusarium solani (Suga et al 2000 Phylogenetic analysis of the phytopathogenic fungus Fusarium solani based on the rDNA-ITS region. Mycol Res 104, 1175), the powdery mildews (Takamatsu et al 1998 Phylogenetic analysis and predicted secondary structures of the rDNA internal transcribed spacers of the powdery mildew fungi (Erysiphaceae). Mycoscience 39, 441), and Basidiomycetes (Sundari et al 2018 Bioprospection of Basidiomycetes and molecular phylogenetic analysis using internal transcribed spacer (ITS) and 5.8 S rRNA gene sequence. Sci Rep 8, 10720).

L110-111: You also need to provide here the methodology on how the data analysis was performed. How long was the ITS sequence used in the multiple sequence alignment?

This information has been integrated in the caption of Fig. 1.

L115: Although you cite a prior paper in the figure caption (58), I think you also need to provide the details on how the analysis was performed in the text because they are not the same. The other manuscript provides an analysis based on two loci as it includes TEF1-α.

More details concerning methodology have been provided.

L121-126: I appreicate that other nuclear/mitochondrial sequences may not be available in Genbank (or other database?) for all species presented here but that doesn't mean that this can be called a phylogenetic analysis. Either acquire the isolates and sequence them, and then include the extra loci in the phylogeny or state that this will need to be done in a future study to verify your gene-tree presented here.

See previous reply to comment concerning line 109.

L204: The meaning of this sentence is unclear, please revise.

L207: Not sure what you mean by "ambivalent aptitude...envisaged". Use 'observed' or 'reported' instead of envisaged. Ambivalent in english means 'unsure' or uncertain, I don't think this is what you mean.

These sentences have been modified to improve clarity and avoid obscure words.

L222: Use 'combat' instead of 'contrast'.

Done.

L234-235: Endophytes do not always provide a benefit to the plant, so please consider revising this sentence to include this point so you don't overstate their potential.

Sentence has been changed in order not to overstate endophytes’ potential.

Reviewer 2:

This is an informative review about two endophytic fungal groups in the Sordariomycetes. I would like to see a bit more endophytic ecology and biology content, since this is about the "endophytism" of the fungi, at least some pictures showing the general endophytic morphology and culture on artificial media would be very helpful for readers to get some ideas about these fungi.

Of course, content of the review basically depends on the available literature, and so far there is no work which has been carried out to depict the development of these fungi within plant tissues.

some minor points:

line 60, what is "organic revision"? not familiar with this term.

‘Comprehensive’ to replace ‘organic’.

2) Bayesian posterior probability higher than 75% should not be considered as "significant", especially for ITS phylogeny

This has been corrected, and Fig. 1 accordingly modified.

Reviewer 2 Report

This is an informative review about two endophytic fungal groups in the Sordariomycetes. I would like to see a bit more endophytic ecology and biology content, since this is about the "endophytism" of the fungi, at least some pictures showing the general endophytic morphology and culture on artificial media would be very helpful for readers to get some ideas about these fungi.

some minor points:

1) line 60, what is "organic revision"? not familiar with this term.

2) Bayesian posterior probability higher than 75% should not be considered as  "significant", especially for ITS phylogeny

Author Response

(The authors gave the same response as above.)

Reviewer 3 Report

This manuscript is a review work and my general opinion of review works is that they have to be performed by researchers with great expertise on the target field. This is my view. In a review, the author has to use his/her expertise to actualize the state of the art of a given research field. Indeed, the own work has to be used; this means that the reference section should include key work of the author/s in the matter.

Unfortunately, this was not the case in the present manuscript, in which the two authors deal with the Endophytism of Lecanicillium and Akanthomyces, without any original contributions in this field other than reviews.

Author Response

(The authors gave the same response as above.)
